# Discovering Order in Unordered Datasets: Generative Markov Networks

## Abstract

The assumption that data samples are independently identically distributed is the backbone of many learning algorithms. Nevertheless, datasets often exhibit rich structures in practice, and we argue that there exist some unknown orders within the data instances. Aiming to find such orders, we introduce a novel Generative Markov Network (GMN) which we use to extract the order of data instances automatically. Specifically, we assume that the instances are sampled from a Markov chain. Our goal is to learn the transitional operator of the chain as well as the generation order by maximizing the generation probability under all possible data permutations. One of our key ideas is to use neural networks as a soft lookup table for approximating the possibly huge, but discrete transition matrix. This strategy allows us to amortize the space complexity with a single model and make the transitional operator generalizable to unseen instances. To ensure the learned Markov chain is ergodic, we propose a greedy batch-wise permutation scheme that allows fast training. Empirically, we evaluate the learned Markov chain by showing that GMNs are able to discover orders among data instances and also perform comparably well to state-of-the-art methods on the one-shot recognition benchmark task.

## 1 Introduction

Recent advances in deep neural networks offer great potentials for machines to learn automatically without humans interventions. For instance, Convolutional Neural Networks (CNNs) (Krizhevsky et al., 2012) provided an automated way for learning image feature representations. Compared to hand-crafted ones such as SIFT and SURF, these hierarchical deep features demonstrate superior performance in recognition (Xu et al., 2015; Finn & Levine, 2017) and transfer learning (Glorot et al., 2011) problems. Another example would be *learning to learn* for automatic parameter estimation. Andrychowicz et al. (2016) proposed to update model parameters without any pre-defined update rule such as stochastic gradient descent (SGD) or ADAM (Kingma & Ba, 2014). Surprisingly, this update-rule-free framework showed better performance and faster convergence on both object recognition and image style transformation tasks. In our paper, we investigate the following novel question: given an unordered dataset where instances may be exhibiting some *implicit* order, can we order a dataset automatically according to this order?

We argue that such order often exists even when we are dealing with the data that are naturally thought of as being i.i.d. sampled from a common though complex distribution. For example, let's consider a dataset consisting of the joint locations on the body of the same person taken on different days. The data i.i.d. assumption is justified since postures of a person took on different days are likely unrelated. However, we can arrange the data instances such that the joints follow an articulated motion or a set of motions in a way that makes each pose highly predictable given the previous ones. Although this arrangement depends on the person as ballerinas' poses might obey different dynamics than the poses of tennis players, the simultaneous inference on the pose dynamics can lead to a robust model that explains the correlations among joints. To put it differently, if we reshuffle the frames of a video clip, the data can now be modeled by an i.i.d. model. Nevertheless, reconstructing the order leads to an alternative model where transitions between the frames are easier to fit the links between the latent structures and observations. The ballerina's dancing, if sampled very sparsely, can be thought of as a reshuffled video sequence that needs to be reordered such that a temporal model can generate it.

One naive and obvious way to find the order in a dataset is to perform sorting based on a predefined distance metric; e.g., the Euclidean distance between image pixel values. However, the distance metrics have to be predefined differently and empirically according to distinct types/characteristics of the datasets at hand. A proper distance metric for one domain may not be a good one for other domains. For instance, $p-$distance is a good measure for DNA/RNA sequences (Nei & Kumar, 2000) while it does not characterize the semantic distances between images. We argue that the key component of the ordering problem lies in the discovery of proper distance metric in an automatic and adaptive way.

To approach this problem, we propose to learn a *distance-metric-free model* to discover the ordering in the dataset. More specifically, we model the data by treating them as if they were generated from a Markov chain. We propose to simultaneously train the transitional operator and find the best order by a joint optimization over the parameter space as well as all possible permutations. We term our model Generative Markov Networks (GMNs). One of the key ideas in the design of GMNs is to use neural networks as a soft lookup table to approximate the possibly huge but discrete transition matrix. This strategy allows GMNs to amortize the space complexity using a unified model. Furthermore, due to the differentiable property of neural networks, the transitional operator of GMNs can also generalize on unseen but similar data instances. As an additional contribution, to ensure the Markov chain learned by GMNs is ergodic, we propose a greedy batch-wise permutation scheme that allows fast training.

One related task is one-shot recognition which has only one labeled data per category in the target domain. Most of the work in this area considered learning a specific distance metric (Koch et al., 2015; Vinyals et al., 2016; Snell et al., 2017) or category-separation metric (Ravi & Larochelle, 2017; Finn et al., 2017) for the data. During the inference phase, they computed either the smallest distance or highest class prediction score between the support and query instances. Alternatively, from a generative modeling perspective, we can first generate the Markov chain for the support instances, then we fit the query instances into the Markov chain and decide the labels with the highest log-likelihood.

Empirically, we evaluate the learned Markov chain by showing that GMNs are able to discover implicit orders among data instances and also perform comparably well to state-of-the-art methods on the benchmark one-shot recognition task.

## 2 RELATED WORK

The literature on deep generative models and stochastic sampling is abundant. Due to the space limit, we discuss the ones that are most relevant to our work.

**Deep Generative Models:** We consider two classes of deep generative models based on *ancestral sampling* and *iterative sampling*, respectively. Variational Autoencoders (VAEs) (Kingma & Welling, 2013) and Generative Adversarial Networks (GANs) (Goodfellow et al., 2014) can be cast as *ancestral sampling*-based methods. In the inference phase, these approaches generated one sample from the model by performing a single inference pass from the underlying graphical models. As a comparison, methods based on *iterative sampling* performed multiple and iterative passes through all the variables in the corresponding graphical models. Usually, these methods involved simulating a Markov chain in the entire state space, and they aimed at improving quality of generated samples by mixing the underlying chain. Recent works on this line of research included (Bengio et al., 2013; 2014; Sohl-Dickstein et al., 2015; Bordes et al., 2017; Song et al., 2017).

Our approach can be categorized as an *iterative sampling*-based model. However, it has three significant differences comparing to previous works. First, all the existing works assumed that training instances are i.i.d. sampled from the stationary distribution of a Markov chain. This assumption is risky since, often the case, it is hard to measure whether a Markov chain has mixed or not. On the contrary, we only assume that data instances are sampled from the chain, without expecting the chain has mixed. As we will see later, the stationarity assumption in previous works often prevents them from observing the implicit data relationships. Second, prior approaches were proposed based on the notion of denoising models. In other words, their goal was generating high-quality images; on the other hand, we aim at discovering orders in datasets. Third, to the best of our knowledge, all the existing works were implicit models in the sense that they only admitted efficient sampling

schemes. In contrast, the proposed GMN is an explicit model where besides an efficient sampling procedure, the model maintains a tractable likelihood function that can be computed efficiently.

**One-Shot Learning:** Deep one-shot learning approaches could be divided into two categories: *distance-metric-learning* and *categories-separation-metric-learning* approaches. The former aimed at either learning a similarity measurement between instance pairs (Koch et al., 2015) or applying specific metric loss based on cosine distance (Vinyals et al., 2016)/ Euclidean distance (Snell et al., 2017). These methods referred to *nonparametric* classifiers and relied heavily upon human design. As a comparison, methods in the second category offered more generalities. Typically, this type of methods tackled the problem using a meta-learning framework to train *parametric* classifiers. Precisely, they considered two levels of learning: the first stage is to update base learners' parameters and the second stage is to update parameters for the meta learner. Recent works (Ravi & Larochelle, 2017; Kaiser et al., 2017; Finn et al., 2017) belonged to this category.

The methods mentioned above viewed one-shot recognition as a discriminative task; on the contrary, we hold a generative perspective. Since we consider a Markov chain data generation assumption, we can directly decide the labels for query instances by fitting them into the Markov chain (or the orders we observe) generated from support instances. This generative nature significantly decreases the difficulty of training as we no longer rely on any designed metric. More details will be covered in Sec. 4.

## 3 GENERATIVE MARKOV NETWORKS

Let $\{s_i\}_{i=1}^n$ denote our training data which are assumed being generated from an unknown Markov chain. Our goal is to jointly recover the unknown Markov chain as well as the order of generation process. Note that since the generation order is unknown, even if the true Markov chain was given, it would still be computationally intractable to find the optimal order that best fits our data. To get around of this intrinsic difficulty, as we will see in Sec. 3.2 , we propose a greedy algorithm to find an order given the current estimation of the transitional operator.

We denote the underlying data order to be a permutation over $[n]$: $\pi = \{\pi(t)\}_{t=1}^n$, where $\pi(t)$ represents the index of the instance that is generated at the $t$-th step of the Markov chain. In other words, a Markov chain is formed as follows:

$$s_{\pi(1)} \to s_{\pi(2)} \to \cdots \to s_{\pi(n)}.$$

We consider all the possible permutations $\pi$ and arbitrary distribution over these permutations, which leads to a joint log-likelihood estimation problem:

$$\max_{\theta,\pi} \log\Big(\sum_\pi \mathcal{P}(\pi)\,\mathcal{P}(\{s_i\}_{i=1}^n, \pi; \theta)\Big) = \max_{\theta,\pi} \log\Big(\sum_\pi \mathcal{P}(\pi)\,\mathcal{P}^{(1)}(s_{\pi(1)})\prod_{t=2}^n \mathcal{T}(s_{\pi(t)}|s_{\pi(t-1)};\theta)\Big),$$

where $\mathcal{P}^{(1)}(\cdot)$ is the initial distribution of the Markov chain and $\mathcal{T}(s'|s;\theta)$ is the transitional operator parametrized by model parameters $\theta$. Note that the effect of the initial distribution $\mathcal{P}^{(1)}(\cdot)$ diminishes with the increase of the data size $n$. Hence, without loss of generality, we assume $\mathcal{P}^{(1)}(\cdot)$ is uniform over all possible states, leading to the following optimization problem:

$$\max_{\theta,\pi} \quad \log\left(\sum_{\pi\in\Pi(n)} \mathcal{P}(\pi)\prod_{t=2}^n \mathcal{T}(s_{\pi(t)}|s_{\pi(t-1)};\theta)\right), \tag{1}$$

where $\Pi(n)$ is the set of all possible permutations over $[n]$. Unfortunately, direct optimization of (1) is computationally intractable. For each fixed $\theta$ and $\mathcal{P}$, the number of all possible permutations (i.e., $|\Pi(n)|$) is $n!$. To approximate this expensive function, we present an efficient greedy algorithm in Sec. 3.3.

### 3.1 PARAMETRIZED TRANSITIONAL OPERATOR VIA NEURAL NETWORKS

In practice, when the state space is huge, often we cannot afford to maintain the tabular transition matrix directly, which takes up to $O(d^2)$ space, where $d$ is the number of states in the chain. For example, if the state refers to a binary image $I \in \{0,1\}^p$, the size of the state space is $d = 2^p$

which is nearly infeasible to compute. Hence, before optimizing (1), we should first find a family of functions to parametrize the transitional operator $\mathcal{T}(\cdot|\cdot)$.

Being universal function approximators (Hornik et al., 1989), neural networks could be used to approximate the discrete structures which led to the recent success of deep reinforcement learning (Mnih et al., 2013). In our case, we utilize neural networks to approximate the discrete tabular transition matrix. The advantages are two-fold: first, it significantly reduces the space complexity by amortizing the space required by each separate state into a unified model. Since all the states share the same model as the transitional operator, there is no need to store the transition vector for each separate state explicitly. Second, neural networks allow better generalization for the transition probabilities across states. The reason is that, in most real-world applications, states, represented as feature vectors, are not independent from each other. As a result, the differentiable approximation to a discrete structure has the additional smoothness properties, which allows the transitional operator to have a good estimate even for the unseen states.

Let $\theta$ be the parameters of the neural networks and we can define

$$f_\theta(s, s') = \mathcal{T}(s'|s; \theta) : \mathbb{R}^p \times \mathbb{R}^p \to [0, 1]$$

to be the transition function that takes two states $s$ and $s'$ as inputs and returns the corresponding transition probability. Note that one can consider each discrete transitional operator as a lookup table; for example, we use $s$ and $s'$ to locate the corresponding row and column of the table and read out its probability. From this perspective, the neural network works as a soft lookup table that outputs the transition probability given two states (features).

### 3.2 GREEDY APPROXIMATION OF THE OPTIMAL ORDER

As mentioned above, the direct evaluation of eq. (1) is computationally intractable given $\mathcal{P}$ and $\theta$. Here, we develop a coordinate ascent style training algorithm to optimize eq. (1) efficiently. The key insight comes from the following observation: for each fixed $\theta$, there exists a point mass distribution over $\Pi(n)$ that achieves the maximum value for eq. (1). More precisely,

$$\max_{\theta, \pi} \ \log \left( \sum_{\pi \in \Pi(n)} \mathcal{P}(\pi) \prod_{t=2}^n \mathcal{T}(s_{\pi(t)}|s_{\pi(t-1)}; \theta) \right) = \max_\theta \ \log \left( \prod_{t=2}^n \mathcal{T}(s_{\pi^*(t)}|s_{\pi^*(t-1)}; \theta) \right)$$

with

$$\pi^* = \arg\max_{\pi \in \Pi(n)} \sum_{t=2}^n \log \mathcal{T}(s_{\pi(t)}|s_{\pi(t-1)}; \theta).$$

We leave the proof in Supplementary. In other words, given each $\theta$, the optimization problem over $\pi$ now reduces to finding the optimal permutation $\pi^*$ that gives the maximum likelihood on generating the data. However, without further assumption on the structure of the transitional operator, this is still a hard problem which takes time $O(n!)$. Instead, we propose a greedy algorithm to approximate the optimal order, which takes time $O(n^2 \log n)$. We list the pseudocode in Alg. 1.

At first, Alg. 1 enumerates all the possible states appearing in the first time step. For each of the following steps, it finds the next state by maximizing the transition probability at the current step, i.e., a local search to find the next state. The final approximate order is then defined to be the maximum of all these $n$ orders. A naive implementation of this algorithm has time complexity $O(n^3)$. However, we can reduce it to $O(n^2 \log n)$ by pre-computing $\mathcal{T}(s_i|s_j; \theta), \forall i, j \in [n]$ and sorting them so that the maximum finding operation in line 5 can be done in constant time.

Given the approximate order $\hat{\pi}$, we then proceed to optimize the model parameter $\theta$ by gradient based optimization. By now it should be clear that the whole algorithm is an instance of the famous coordinate ascent algorithm, where we alternatively optimize over the order $\pi$ and the model parameters $\theta$. Since both optimizations over $\theta$ and $\pi$ will not decrease the objective function, the algorithm is guaranteed to converge.

### 3.3 BATCH-WISE PERMUTATION TRAINING

The $O(n^2 \log n)$ computation to find the approximate order in Alg. 1 can be expensive when the size of the data is large. In this section we provide batch-wise permutation training to avoid this

---

**Algorithm 1** Greedy Approximate Order

---

**Input:** Input data $\{s_i\}_{i=1}^n$ and transitional operator $\mathcal{T}(s_i|s_j;\theta)$
1: $v^* \leftarrow -\infty$
2: **for** $i = 1$ to $n$ **do**
3:      $\pi_i(1) \leftarrow i$
4:      **for** $j = 2$ to $n$ **do**
5:          $\pi_i(j) \leftarrow \max_{k \notin \{\pi_i(1),...,\pi_i(j-1)\}} \mathcal{T}(s_k \mid s_{\pi_i(j-1)};\theta)$
6:      **end for**
7:      $v_i \leftarrow \sum_{t=2}^n \log \mathcal{T}(s_{\pi_i(j-1)}|s_{\pi_i(t)};\theta)$
8:      **if** $v_i > v^*$ **then**
9:          $\hat{\pi} \leftarrow \pi_i$
10:         $v^* \leftarrow v_i$
11:      **end if**
12: **end for**
13: **return** $\hat{\pi}$

---

**Algorithm 2** Optimization with Batch-Wise Permutation Training

---

**Input:** $\{s_i\}_{i=1}^n, b_o, b, t, \gamma$
1: Initialize $\theta^{(0)}, \{x_i^{(0)}\}_{i=1}^b$
2: **for** $k = 1$ to $\infty$ **do**
3:      **if** $k \equiv 1(\mathrm{mod}\ t)$ **then**
4:          Sample $\{x_i^{(k)}\}_{i=1}^{b-b_o} \sim \{s_i\}_{i=1}^n$
5:          $\{x_i^{(k)}\}_{i=1}^b = \{\{x_i^{(k)}\}_{i=1}^{b-b_o}, \{x_i^{(k-1)}\}_{i=1}^{b_o}\}$
6:      **end if**
7:      Compute $\hat{\pi}^{(k)}$ using the Greedy Approximate Order (Alg. 1)
8:      Compute $\nabla_\theta^{(k-1)} \log \mathcal{P}(\{x_i\}_{i=1}^b; \theta^{(k-1)}) = \partial_\theta \sum_{t=2}^b \log \mathcal{T}(x_{\hat{\pi}^{(k)}(t)}|x_{\hat{\pi}^{(k-1)}(t-1)}; \theta^{(k-1)})$
9:      $\theta^{(k)} = \theta^{(k-1)} + \gamma \nabla_\theta^{(k-1)} \log \mathcal{P}(\{x_i\}_{i=1}^b; \theta^{(k-1)})$
10: **end for**

---

issue. The idea is to partition the original training set into batches with size $b$ and perform greedy approximate order on each batch. Assuming $b \ll n$ is a constant, the effective time complexity becomes: $O(b^2 \log b) \cdot n/b = O(nb \log b)$, which is linear in $n$.

However, since training data are partitioned into chunks, the learned transitional operator is not guaranteed to have nonzero transition probabilities between different chunks of data. In other words, the learned transitional operator does not necessarily induce an ergodic Markov chain due to the isolated states. To avoid this problem, we propose a simple strategy to enforce some samples are overlapping between the consecutive batches. We show the pseudocode in Alg. 2. In Alg. 2, $b$ means the batch size, $\gamma$ is the learning rate and $b_0 < b$ is the number of overlap states between consecutive batches.

## 3.4 Introducing Stochastic Latent Variables via Variational Bayes Inference

In this section we give a detailed description on how to implement the transitional operator where the state can be both discrete or continuous. At the first step, to prevent our GMNs from simply memorizing all the training data and their transitions, we introduce stochastic latent variables $z \in \mathbb{R}^\mathbf{z}$ via Variational Bayes Inference (Wainwright et al., 2008). The evidence lower bound (ELBO) of the log likelihood for the transitional operator (i.e., $\log \mathcal{T}(s'|s;\theta)$) becomes:

$$\log \mathcal{T}(s'|s;\theta) \approx \ \mathbf{E}_{z \sim \mathcal{Q}(z|s;\phi)}\Big[\log \mathcal{P}(s'|s,z;\psi)\Big] - \mathcal{KL}\Big(\mathcal{Q}(z|s;\phi) \,||\, \mathcal{P}(z)\Big), \qquad (2)$$

where $\mathcal{T}(s'|s;\theta)$ has been replaced by a distribution $\mathcal{P}(s'|s,z;\psi)$ parametrized by $\psi$, which allows us to make the dependence of $s$ on $z$. Moreover, $\mathcal{KL}$ is the KL-divergence, $\mathcal{Q}(z|s;\phi)$ is an encoder function parametrized by $\phi$ that encodes latent code $z$ given current state $s$, and $\mathcal{P}(z)$ is a fixed prior which we take its form as Gaussian distribution $\mathcal{N}(0, \mathbf{I})$. We use reparametrized trick to draw $\mathcal{Q}(z|s;\phi)$ from Gaussian $\mathcal{N}\big(\mu_{\mathcal{Q},\phi}(s), \sigma_{\mathcal{Q},\phi}^2(s)\mathbf{I}\big)$ where $\mu_{\mathcal{Q},\phi}(s)$ and $\sigma_{\mathcal{Q},\phi}(s)$ are learnable functions.

Next, we consider two types of distribution family for $\mathcal{P}(s'|s,z;\theta)$: Bernoulli and Gaussian.

Figure 1: For MNIST, Horse, and MSR_SenseCam datasets: the implicit order observed from GMN and the oder implied from Nearest Neighbor sorting.

If $s \in \{0,1\}^p$ (i.e., a binary image), we define $\log \mathcal{P}(s'|s,z;\psi)$ as:

$$\log \mathcal{P}(s'|s,z;\psi) = s' \odot \log\Big(g_\psi(s,z)\Big) + (1-s') \odot \log\Big(1 - g_\psi(s,z)\Big),$$

where $\odot$ is element-wise multiplication and $g_\psi(s,z) : \Big[\{0,1\}^p, \mathbb{R}^{\mathbf{z}}\Big] \to [0,1]^p$.

If $s \in \mathbb{R}^p$ (i.e., a real-valued feature vector), we choose $\mathcal{P}(s'|s,z;\psi)$ to be fixed variance factored Gaussian $\mathcal{N}\Big(\mu_{\mathcal{P},\psi}(s,z), \sigma_{\mathcal{P}}^2 \mathbf{I}\Big)$, where $\mu_{\mathcal{P},\psi}(s,z) : \mathbb{R}^{p+\mathbf{z}} \to \mathbb{R}^p$ and $\sigma_{\mathcal{P}}$ is a fixed variance. We simply choose $\sigma_{\mathcal{P}}$ in all the experiments. $\log \mathcal{P}(s'|s,z;\theta)$ can thus be defined as

$$\log \mathcal{P}(s'|s,z;\psi) = -\frac{1}{2\sigma_{\mathcal{P}}^2}||s' - \mu_{\mathcal{P},\psi}(s,z)||_2^2 + const.,$$

where $const.$ is not related to the optimization of $\psi$.

For simplicity, we specify $\theta = \{\psi \cup \phi\}$. Therefore, the model parameters update for $\theta$ in (2) refers to the updates for $\psi$ and $\phi$.

## 4 EXPERIMENTS

### 4.1 DISCOVERING ORDERS IN DATASETS

We perform experiments on ordering data in three datasets: MNIST (LeCun et al., 1990), Horse (Borenstein & Ullman, 2002), and MSR_SenseCam (Jojic et al., 2010). We also provide another experiment on Moving MNIST (Srivastava et al., 2015) in Supplementary. Among these datasets, MNIST, Horse, and MSR_SenseCam do not have explicit orders. On the other hand, Moving MNIST can be seen as a collection of short video clips, and thus each sequence of frames has an explicit order. Due to the space limit, we only show partial ordering results. Please see Supplementary for the full version.

<**MNIST**> MNIST (LeCun et al., 1990) is a well-studied dataset that contains 60,000 training examples. Each example is a digit image with size 28x28. We rescale the pixel values to $[0,1]$. Note that since MNIST contains a large number of instances, we perform the ordering in a randomly sampled batch to demonstrate our results.

<**Horse**> Horse dataset (Borenstein & Ullman, 2002) consists of 328 horse images collected from the Internet. Each horse is centered in a 30x40 image. For the preprocessing, the object-background segmentation is applied, and the binary pixel value is set to $1$ and $0$ for object and background, respectively. Examples are show in Supplementary.

<**MSR_SenseCam**> MSR_SenseCam (Jojic et al., 2010) is a dataset consisting of images taken by SenseCam wearable camera. It contains 45 classes with approximately 150 images per class. Each image has size 480x640. We resize each image into 224x224 and extract the feature from VGG-19 network (Simonyan & Zisserman, 2014). In this dataset, we consider only *office* category which has 362 images.

### 4.1.1 IMPLICIT ORDERS IN DATASETS

We apply Alg. 2 to train our Generative Markov Networks. When the training converges, we plot the images following permutation $\hat{\pi}$ in Alg. 1. Note that $\hat{\pi}$ can be seen as the implicit order suggested by GMNs. For comparison, we also plot the images following nearest neighbor sorting using

Figure 2: For MNIST, Horse, and MSR_SenseCam datasets: data generation from the learned transitional operator in GMN and Nearest Neighbor search.

Euclidean distances. The parameters $\{b_{overlap}, b, t\}$ in Alg. 2 are $\{50, 500, 600\}$, $\{328, 328, 1\}$, and $\{362, 362, 1\}$ for MNIST, Horse, and MSR_SenseCam, respectively. Network architectures for parameterizing $\mathcal{T}(\cdot|\cdot; \theta)$ are specified in Supplementary.

The results are shown in Fig. 1. We first observe that data following the order suggested by our proposed GMN have visually high autocorrelation. This result implies that our proposed GMN can discover nice implicit orders for the dataset. Comparing to the strong ordering baseline Nearest Neighbor sorting, one could hardly tell which one is better. Nevertheless, GMN is a *distance-metric-free* model which requires no predefined distance metric. Moreover, the implicit order suggested by GMN considers a generative modeling viewpoint: the order is the optimal permutation under the Markov chain data generation assumption (see Sec. 3.2).

### 4.1.2 TRANSITIONAL OPERATOR AS A GENERATIVE MODEL

Next, we examine the data generation using the learned transitional operator. Conditioned on a given sample $s$, instead of sampling $s' \sim \mathcal{T}(s'|s; \theta)$ directly, we sample $s' = \arg\max_{s^\dagger \in \{\{s_i\}_{i=1}^n \setminus s\}} \mathcal{T}(s^\dagger|s; \theta)$. We make this modification based on the reason that our model aims at discovering datasets' orders, while other *iterative sampling* models (Bengio et al., 2013; 2014; Sohl-Dickstein et al., 2015; Bordes et al., 2017; Song et al., 2017) intended to denoise generated samples. Similar to Sec. 4.1.1, we exploit nearest neighbor search using Euclidean distance for comparison. More precisely, $s'_{NN} = \arg\max_{s^\dagger \in \{\{s_i\}_{i=1}^n \setminus s\}} d(s^\dagger, s)$ with $d(\cdot, \cdot)$ denoting Euclidean distance.

Fig. 2 illustrates the sampling of GMN and Nearest Neighbor search. We can see that Nearest Neighbor search is not able to perform efficient sampling since it would stick between two similar images. On the other hand, our proposed GMN can perform consecutive sampling. This tremendous difference implies the distinction between the discriminative (sampling by a fixed distance metric) and the generative (sampling through the transitional operator in a Markov chain) model.

### 4.2 ONE-SHOT RECOGNITION

Now, we perform one-shot recognition task on the miniImageNet (Vinyals et al., 2016; Ravi & Larochelle, 2017), which is a benchmark dataset designed for the evaluation of few-shot learning (Vinyals et al., 2016; Ravi & Larochelle, 2017). Being a subset of ImageNet (Russakovsky et al., 2015), it contains 100 classes and each class has 600 images. Each image is downsampled to size 84x84. As suggested in (Ravi & Larochelle, 2017), the dataset is divided into three parts: 64 classes for training, 16 classes for validation, and 20 classes for testing. Identical to (Ravi & Larochelle, 2017), we consider the $5-$way $1-$shot problem. That is, from testing classes, we sample 5 classes with each class containing 1 labeled example. The labeled examples refer to support instances. Then, we randomly sample 500 unlabeled query examples in these 5 classes for evaluation. We repeat this procedure for $10,000$ times and report the average with $95\%$ confidence intervals in Tbl. 1.

### 4.2.1 TRAINING DETAILS

Instead of viewing one-shot recognition as a discriminative task, we hold it as a generative one. To achieve this goal, we train our Generative Markov Networks on training classes and then apply it to testing classes. More precisely, for each training episode, we sample 1 class from the training

Table 1: 5-way 1-shot recognition task for miniImageNet. The results are reported averagely in $10,000$ episodes with $95\%$ confidence intervals.

| Model | Basic/ Advanced Model | Discriminative/ Generative | Parametric/ Nonparametric | Accuracy |
|---|---|---|---|---|
| Meta-Learner LSTM (Ravi & Larochelle, 2017) | Basic | Discriminative | Parametric | 43.44±0.77 |
| Model-Agnostic Meta-Learning (Finn et al., 2017) | Basic | Discriminative | Parametric | 48.70±1.84 |
| Meta Networks (Munkhdalai & Yu, 2017) | Advanced | Discriminative | Parametric | 49.21±0.96 |
| Meta-SGD (Li et al., 2017) | Basic | Discriminative | Parametric | 50.47±1.87 |
| Temporal Convolutions Meta-Learning (Mishra et al., 2017) | Advanced | Discriminative | Parametric | 55.71±0.99 |
| Nearest Neighbor with Cosine Distance | Basic | Discriminative | Nonparametric | 41.08±0.70 |
| Matching Networks FCE (Vinyals et al., 2016) | Basic | Discriminative | Nonparametric | 43.56±0.84 |
| Siamese (Koch et al., 2015) | Basic | Discriminative | Nonparametric | 48.42±0.79 |
| mAP-Direct Loss Minimization (Triantafillou et al., 2017) | Basic | Discriminative | Nonparametric | 41.64±0.78 |
| mAP-Structural Support Vector Machine (Triantafillou et al., 2017) | Basic | Discriminative | Nonparametric | 47.89±0.78 |
| Prototypical Networks (Snell et al., 2017) | Basic | Discriminative | Nonparametric | 49.42±0.78 |
| Attentive Recurrent Comparators (Shyam et al., 2017) | Not Specified | Discriminative | Nonparametric | 49.1 |
| Skip-Residual Pairwise Networks (Mehrotra & Dukkipati, 2017) | Advanced | Discriminative | Nonparametric | 55.2 |
| Generative Markov Networks without fine-tuning (ours) | Basic | Generative | Nonparametric | 45.36±0.94 |
| Generative Markov Networks with fine-tuning (ours) | Basic | Generative | Nonparametric | 48.87±1.10 |

classes and let $\{s_i\}_{i=1}^n$ be all the data from this class. Then, we apply Alg. 2 with $\{b_{overlap}, b, t\} = \{20, 100, 10\}$. We consider $3,000$ training episodes.

On the other hand, for each testing episode, we apply GMNs to generate a chain from each support instance:

$$\tilde{s}_1^c \sim \mathcal{T}(\cdot|s_0^c; \theta), \ \tilde{s}_2^c \sim \mathcal{T}(\cdot|\tilde{s}_1^c; \theta), \ \cdots, \tilde{s}_k^c \sim \mathcal{T}(\cdot|\tilde{s}_{k-1}^c; \theta),$$

where $s_0^c$ is the support instance belonging to class $c$ and $\tilde{s}^c$ is the generated samples from the Markov chain.

Next, we fit each query example into each chain by computing the average approximating log-likelihood. Namely, the probability for generating the query sample $s_q$ in the chain of class $c$ is

$$\mathcal{P}(s_q|c) := \frac{1}{k+1} \left( \log \mathcal{T}(s_q|s_0^c; \theta) + \sum_{i=1}^{k} \log \mathcal{T}(s_q|\tilde{s}_i^c; \theta) \right). \tag{3}$$

In a generative viewpoint, the predicted class $\hat{c}$ for $s_q$ is determined by

$$\hat{c} = \arg\max_c \ \mathcal{P}(s_q|c). \tag{4}$$

For fair comparisons, we use the same architecture specified in (Ravi & Larochelle, 2017) to extract 1600-dimensional features. We pretrain the architecture using standard softmax regression on image-label pairs in training and validation classes. The architecture consists of 4 blocks. Each block comprises a CNN layer with 64 3x3 convolutional filters, Batch Normalization (Ioffe & Szegedy, 2015) layer, ReLU activation, and 2x2 Max-Pooling layer. Then, we train our Generative Markov Networks based on these $1,600$ dimensional features. Network architecture for parameterizing $\mathcal{T}(\cdot|\cdot; \theta)$ is specified in Supplementary.

### 4.2.2 RESULTS

For a comprehensive analysis, we also provide the variant of our GMN with fine-tuning. In other words, we fine-tune GMN by applying Alg. 2 with $\{b_{overlap}, b, t\} = \{20, 100, 10\}$ on support and query instances. Note that in eq. (3), $k$ is chosen to be 1 and 5 for the non-fine-tuned and fine-tuned version, respectively. We compare our proposed method and the related approaches in Tbl. 1, in which *Basic* model refers the architecture in (Ravi & Larochelle, 2017) and *Advanced* models refer to more complicated designs. Generally, it is not fair to compare the methods using different models; therefore, we only discuss the methods using *Basic* model in the following.

First, we observe that the performance of GMN is comparable to other works. For example, the best result of all methods is reported by Meta-SGD (Munkhdalai & Yu, 2017) with $50.47 \pm 1.87$. Although GMN suffers from slight performance drop, it requires a much less computational budget. The reason is that the meta-learning (parametric) approaches (Ravi & Larochelle, 2017; Finn et al., 2017; Munkhdalai & Yu, 2017; Li et al., 2017; Mishra et al., 2017) rely on huge networks to manage complicated intersections between meta and base learners, while parameters for GMN exist only in $\theta$ which is a relatively tiny network. On the other hand, the best performance reported in the distance-metric learning (nonparametric) approaches is Prototypical Networks (Snell et al., 2017) with $49.42 \pm 0.78$. Sacrificing from little performance deterioration, our proposed GMN enjoys

more flexibility without the need of defining any distance metric as in (Vinyals et al., 2016; Koch et al., 2015; Triantafillou et al., 2017; Snell et al., 2017; Shyam et al., 2017; Mehrotra & Dukkipati, 2017). More importantly, except for our proposed GMN, all the works belong to discriminative models, which means they are optimized based on carefully chosen objectives for one-shot learning purpose.

Next, our proposed GMN enjoys a significant improvement ($45.36 \pm 0.94 \rightarrow 48.87 \pm 1.10$) from fine-tuning over support and query instances. This result verifies that GMN is able to simulate the Markov chain data generation process since the query instances can be better fitted in the chains generated from the support instances.

## 5  CONCLUSION

In this paper, we argue that data i.i.d. assumption is not always the case in most of the datasets. Often, data instances are exhibiting some implicit orders which may benefit our understanding and analysis of the dataset. To observe the implicit orders, we propose a novel Generative Markov Network which considers a Markov chain data generation scheme. Specifically, we simultaneously learn the transitional operator as a generative model in the Markov chain as well as find the optimal orders of the data under all possible permutations. In lots of experiments, we show that our model is able to observe implicit orders from unordered datasets and also perform well on the one-shot recognition task.

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

# SUPPLEMENTARY FOR
# DISCOVERING ORDER IN UNORDERED DATASETS: GENERATIVE MARKOV NETWORKS

**Anonymous authors**

## 1 PROOF FOR SEC. 3.2

Here, we prove that

$$\max_{\theta,\pi} \log\left(\sum_{\pi\in\Pi(n)} \mathcal{P}(\pi)\prod_{t=2}^{n}\mathcal{T}(s_{\pi(t)}|s_{\pi(t-1)};\theta)\right) = \max_{\theta} \log\left(\prod_{t=2}^{n}\mathcal{T}(s_{\pi^*(t)}|s_{\pi^*(t-1)};\theta)\right)$$

with

$$\pi^* = \arg\max_{\pi\in\Pi(n)} \sum_{t=2}^{n} \log \mathcal{T}(s_{\pi(t)}|s_{\pi(t-1)};\theta).$$

*Proof:*

For any $\pi$ over $\Pi(n)$, we have:

$$\log\left(\sum_{\pi} \mathcal{P}(\pi)\prod_{t=2}^{n}\mathcal{T}(s_{\pi(t)}|s_{\pi(t-1)};\theta)\right) \leq \log\left(\sum_{\pi} \mathcal{P}(\pi)\prod_{t=2}^{n}\mathcal{T}(s_{\pi^*(t)}|s_{\pi^*(t-1)};\theta)\right)$$

$$= \log\left(\sum_{\pi} \mathbb{I}(\pi=\pi^*)\prod_{t=2}^{n}\mathcal{T}(s_{\pi^*(t)}|s_{\pi^*(t-1)};\theta)\right)$$

$$= \log\left(\prod_{t=2}^{n}\mathcal{T}(s_{\pi^*(t)}|s_{\pi^*(t-1)};\theta)\right),$$

where $\mathbb{I}(\pi=\pi^*)$ is the indicator function that takes value 1 iff $\pi=\pi^*$ otherwise 0. Realize that $\mathbb{I}(\pi=\pi^*)$ also defines a valid distribution over $\Pi(n)$, which proves our claim.

## 2 EXAMPLE IMAGES FOR HORSE DATASET

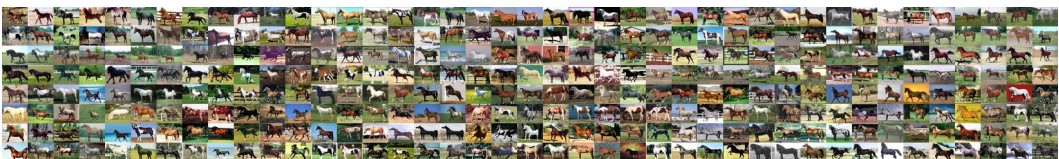

Figure 1: RGB images of Horse Dataset.

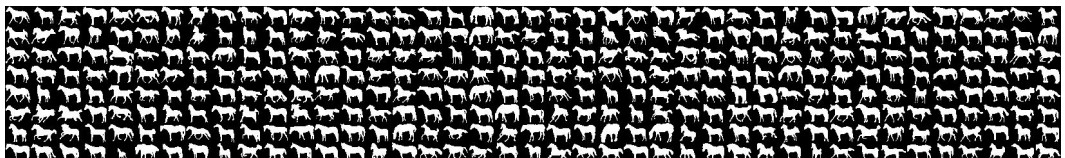

Figure 2: Pre-processed images of Horse Dataset.

Horse dataset Borenstein & Ullman (2002) consists of images collected from Internet. Fig. 1 illustrates these RGB images. Object-background segmentations are applied on these images and the horses are centered in 30x40 images. The processed images are shown in Fig. 2.

# 3 FULL ORDERING RESULTS FOR MNIST, HORSE, AND MSR_SENSECAM

Fig. 3, 4, and 5 show the results of the implicit order observed from GMN the order implied from Nearest Neighbor sorting. On the other hand, Fig. 6, 7, and 8 illustrate the sampling of GMN and Nearest Neighbor search.

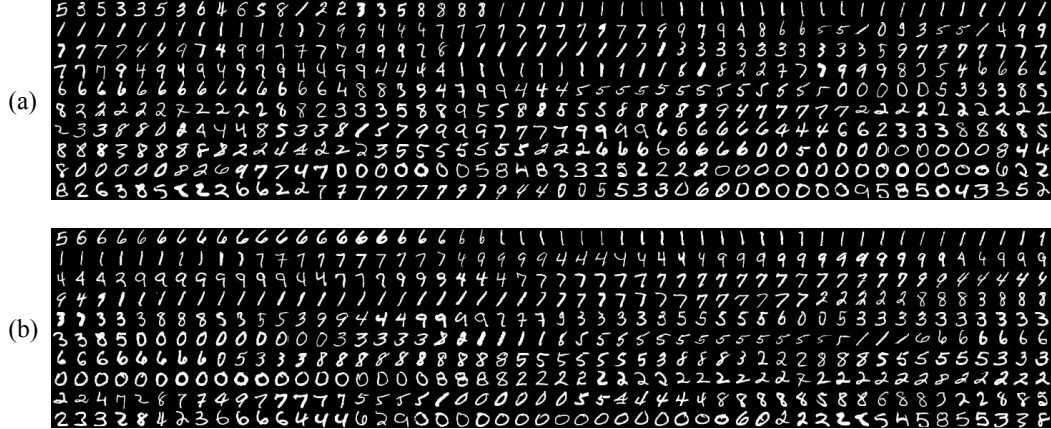

Figure 3: For MNIST dataset: (a) implicit order observed from GMNs (b) order implied from nearest neighbor sorting using Euclidean distance.

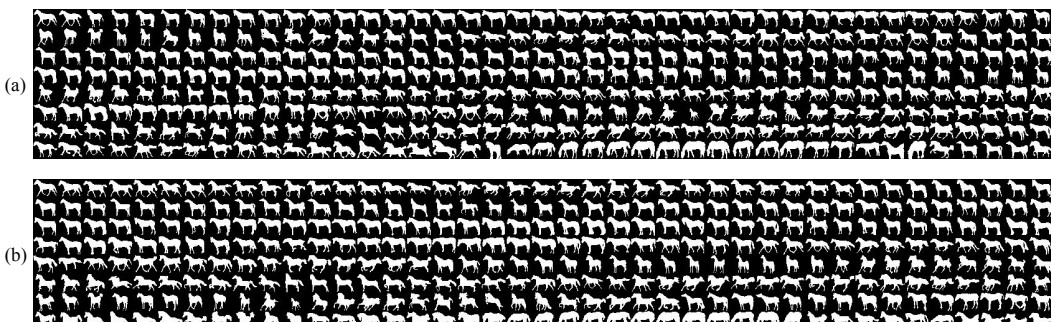

Figure 4: For Horse dataset: (a) implicit order observed from GMNs (b) order implied from nearest neighbor sorting using Euclidean distance.

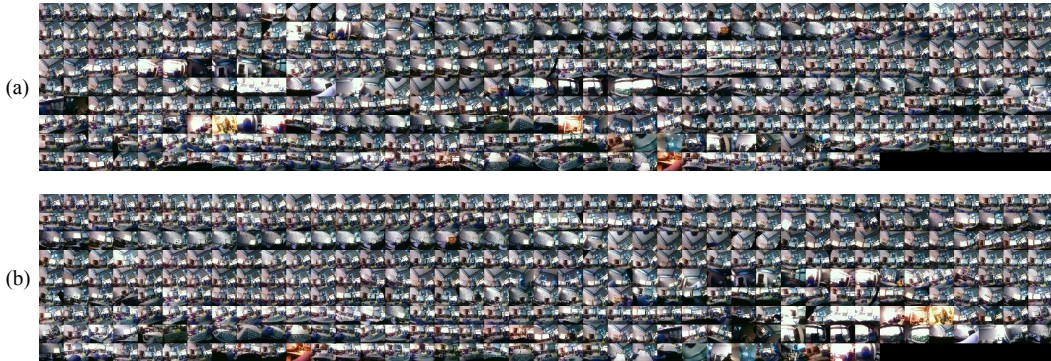

Figure 5: For *office* category in SenseCam dataset: (a) implicit order observed from GMNs (b) order implied from nearest neighbor sorting using Euclidean distance.

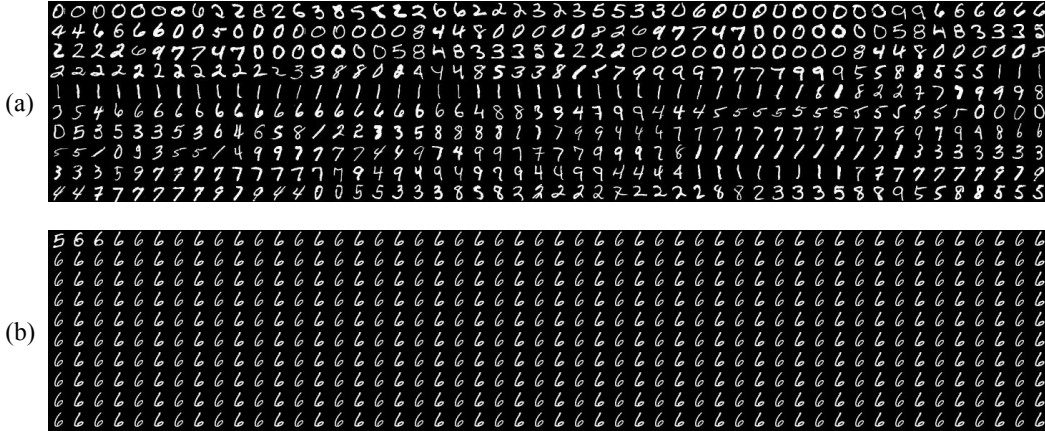

Figure 6: For MNIST dataset, data generation from (a) learned transition operator in GMNs (b) nearest neighbor search using Euclidean distance.

## 4 MOVING MNIST

<**Moving MNIST**> Moving MNIST (Srivastava et al., 2015) contains $10,000$ sequences each of length 20 showing 2 digits moving in a 64x64 frame. We rescale the pixel values to $[0,1]$. For each training episode, we apply Alg. 2 to train GMN on one randomly chosen sequence with parameters $\{b_{overlap}, b, t\}$ set as $\{0, 20, 10\}$. We consider $6,000$ training episodes. For evaluation, we randomly sample a disjoint sequence from training sequences and observe the optimal permutation (implicit order) from Alg. 1.

Fig. 9 illustrates the results for the implicit order observed from Generative Markov Networks, the order inferred from Nearest Neighbor sorting using Euclidean distance, and the suggested explicit order. We find that both the orders observed from GMN and NN sorting manifest smooth motions for two digits in the frame. It is worth noting that our proposed GMN enjoys the freedom of not defining any distance metric.

The sampling results for Moving MNIST dataset are shown in Fig. 10. We consider two approaches: the proposed Generative Markov Networks and Nearest Neighbor search. We find that, by learning the transition operator in a Markov chain as a generative model, GMN performs much better sampling results than Nearest Neighbor search which is a discriminative model.

## 5 NETWORK ARCHITECTURES FOR TRANSITION OPERATOR

We elaborate the design of the transition operator in Fig. 11. In our design, $U$ can be seen as a gating mechanism between input $X_t$ and the learned update $\tilde{X}$. More precisely, the output can be written

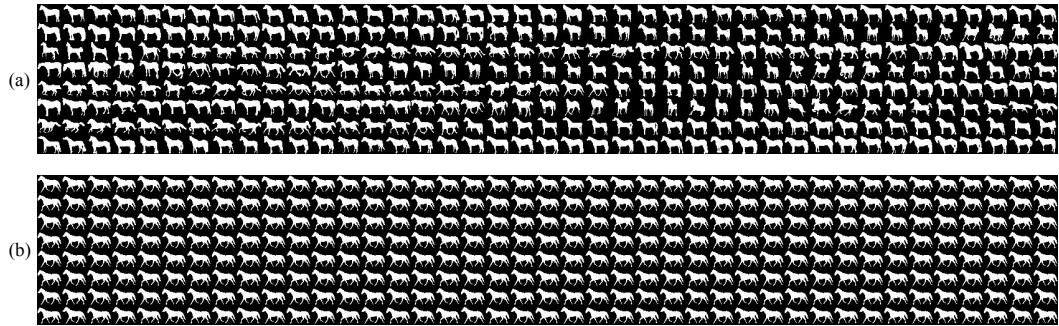

Figure 7: For Horse dataset, data generation from (a) learned transition operator in GMNs (b) nearest neighbor search using Euclidean distance.

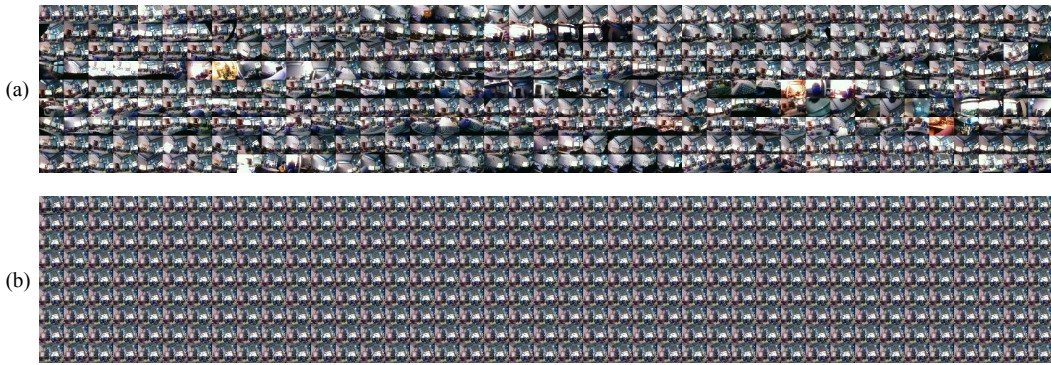

Figure 8: For *office* category in SenseCam dataset, data generation from (a) learned transition operator in GMNs (b) nearest neighbor search using Euclidean distance.

as

$$X_{t+1} = U \odot \tilde{X} + (\mathbf{1} - U) \odot X_t, \tag{1}$$

where $\odot$ denotes element-wise product. We specify each function $f$ in Tbl. 1, 2, 3, 4, and 5. Note that we omit the bias term for simplicity. We use ADAM (Kingma & Ba, 2014) with learning rate 0.001 and 0.2 dropout rate to train our $\mathcal{T}(\cdot|\cdot;\theta)$.

Table 1: Details of functions for MNIST experiments.

| function | details |
|---|---|
| f1 | 784x512 FC layer with ReLU |
| f21 | 512x128 FC layer |
| f22 | 512x128 FC layer |
| f3 | 912x512 FC layer with ReLU |
| f41 | 512x784 FC layer with sigmoid |
| f42 | 512x784 FC layer with sigmoid |

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

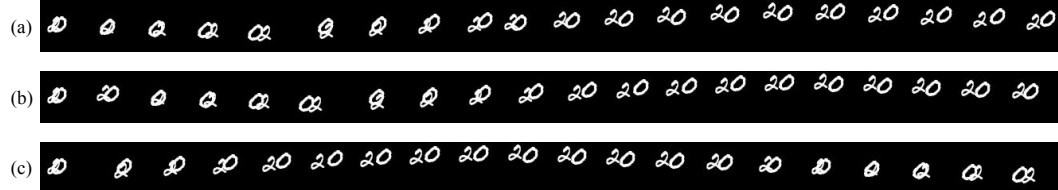

Figure 9: For Moving MNIST dataset: (a) implicit order observed from GMNs (b) order implied from nearest neighbor sorting using Euclidean distance (c) suggested explicit order.

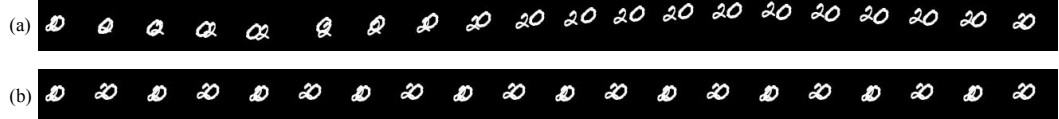

Figure 10: For Moving MNIST dataset, data generation from (a) learned transition operator in GMNs (b) nearest neighbor search using Euclidean distance.

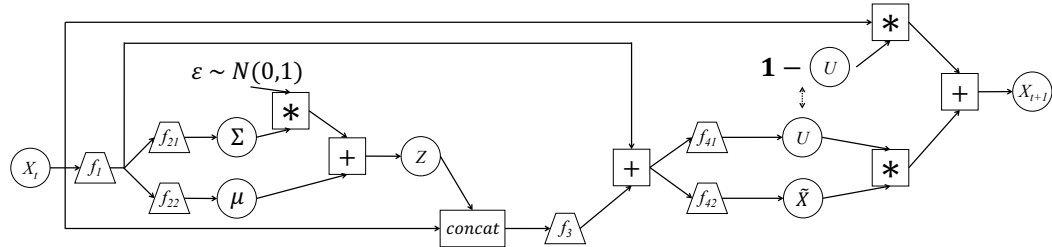

Figure 11: Network design for $\mathcal{T}(\cdot|\cdot;\theta)$.

Table 2: Details of functions for Horse experiments.

| function | details |
|---|---|
| f1 | 1200x512 FC layer with ReLU |
| f21 | 512x128 FC layer |
| f22 | 512x128 FC layer |
| f3 | 912x512 FC layer with ReLU |
| f41 | 512x1200 FC layer with sigmoid |
| f42 | 512x1200 FC layer with sigmoid |

Table 3: Details of functions for MSR_SenseCam experiments.

| function | details |
|---|---|
| f1 | 4096x1024 FC layer with ReLU |
| f21 | 1024x256 FC layer |
| f22 | 1024x256 FC layer |
| f3 | 4352x1024 FC layer with ReLU |
| f41 | 1024x4096 FC layer with sigmoid |
| f42 | 1024x4096 FC layer |

Table 4: Details of functions for Moving MNIST experiments.

| function | details |
|---|---|
| f1 | 4096x1024 FC layer with ReLU |
| f21 | 1024x256 FC layer |
| f22 | 1024x256 FC layer |
| f3 | 4352x1024 FC layer with ReLU |
| f41 | 1024x4096 FC layer with sigmoid |
| f42 | 1024x4096 FC layer with sigmoid |

Table 5: Details of functions for miniImageNet experiments.

| function | details |
| --- | --- |
| f1 | 1600x1024 FC layer with ReLU // 1024x512 FC layer with ReLU // 512x256 FC layer with ReLU |
| f21 | 256x64 FC layer |
| f22 | 256x64 FC layer |
| f3 | 1664x256 FC layer with ReLU |
| f41 | 256x512 FC layer with ReLU // 512x1024 FC layer with ReLU // 1024x1600 FC layer with sigmoid |
| f42 | 256x512 FC layer with ReLU // 512x1024 FC layer with ReLU // 1024x1600 FC layer |

