# OpenReview forum: "Discovering Order in Unordered Datasets: Generative Markov Networks"
_ICLR.cc/2018/Conference — Reject_

### Official Review · AnonReviewer1 · 2017-11-28
**[updated] Reject - interesting ideas, but weak presentation and experiments**

**Rating:** 4
**Confidence:** 4

**Review:**

[After rebuttal]:
I appreciate the effort the authors have put into the rebuttal, but I do not see a paper revision or new results, so I keep my rating.

---

The paper proposes “Generative Markov Networks” - a deep-learning-based approach to modeling sequences and discovering order in datasets. The key ingredient of the model is a deep network playing the role of a transition operator in Markov chain, trained via Variational Bayes, similar to a variational autoencoder (but with non-identical input and output images). Given an unordered dataset, the authors maximize its likelihood under the model by alternating gradient ascent steps on the parameters of the network and greedy reordering of the dataset. The model learns to find reasonable order in unordered datasets, and achieves non-trivial performance on one-shot learning.

Pros:
1) The one-shot learning results are promising. The method is conceptually more attractive than many competitors, because it does not involve specialized training on the one-shot classification task. The ability to perform unsupervised fine-tuning on the target test set is also appealing.
2) The idea of explicitly representing the neighborhood structure within a dataset is generally interesting and seems related to the concept of low-dimensional image manifold. It’s unclear why does this manifold have to be 1-dimensional, though.

Cons:
1) The motivation of the paper is not convincing. Why does one need to find order in unordered datasets? The authors do not really discuss this at all, even though this seems to be the key task in the paper, as reflected in the title. What does one do with this order? How does one even evaluate if a discovered order is good or not?
2) The one-shot classification results are to me the strongest part of the paper. However, they are rushed and not analyzed in detail. It is unclear which components of the system contribute to the performance. As I understand the method, the authors effectively select several neighbors of the labeled samples and then classify the remaining samples based on the average similarity to these. What if the same procedure is performed with a different similarity measure, not the one learned by GMN? I am not convinced that the proposed method is well tuned for the task. Why is it useful to discover one-dimensional structure, rather than learning a clustering or a metric? Could it be that with a different similarity measure (like the distance in the feature space of a network trained on classification) this procedure would work even better? Or is GMN especially good for this task? If so. why?
3) The experiments on dataset ordering are not convincing. What should one learn from those? There are no quantitative results, just a few examples (and more in the supplement). The authors even admit that “Comparing to the strong ordering baseline Nearest Neighbor sorting, one could hardly tell which one is better”. Nearest neighbor with Euclidean metric is not a strong baseline at all, and not being able to tell if the proposed method is better than that is not a good sign.
4) The authors call their method distance-metric-free. This is strange to me. The loss function used during training of the network is a measure of similarity between two samples (may or may not be a proper distance metric). So the authors do assume having some similarity measure between the data points. The distance-metric-free claim is similar to saying that negative log-likelihood of a Gaussian has nothing to do with Euclidean distance.
5) The experiments on using the proposed model as a generative model are confusing. First, the authors do not generate the samples directly, but instead select them from the dataset - this is quite unconventional. Then, the NN baseline is obviously doomed to jump between two samples - the authors could come up with a better baseline, for instance linearly extrapolating based on two most recent samples, or learning the transition operator with a simple linear model.
6) I am puzzled by the hyperparameter choices. It seems there was a lot of tuning behind the scenes, and it should be commented on. The parameters are very different between the datasets (top of page 7), why is that? Why do they have to differ so much - is the method very unstable w.r.t. the parameters? How can it be that b_{overlap} = b ? Also, in the one-shot classification results, the number of sampled neighbors is 1 without fine-tuning and 5 with fine-tuning - this is strange and not explained.
7) This work seems related to simultaneous clustering and representation learning, in that it combines discrete reordering and continuous deep network training. The authors should perhaps mention this line of work. See e.g. Yang et al. “Joint Unsupervised Learning of Deep Representations and Image Clusters”, CVPR 2016.

To conclude, the paper has some interesting ideas, but the presentation is not convincing, and the experiments are substandard. Therefore at this point I cannot recommend the paper for publication.

---

> ### Author Response · Authors · 2017-12-22
> **Rebuttal**
>
> We thank the Reviewer for pointing out the possible improvements on the paper.
>
> 1. [Concerns on the Motivation and Quantitative Results]
>
> Consider the task of studying evolutions for galaxy or star systems. Usually, the process takes millions or even billions of years, and it is infeasible for a human to collect successive data points manifesting meaningful changes. Therefore, we propose to recover the evolution when just providing a snapshot of thousands of data points. Similar arguments can be made in the study of slow-moving human diseases such as Parkinson's. On the opposite side, the cellular or molecular processes are too fast to permit entire trajectories. In these applications, scientists would like to recover the order from non-sequenced and individual data, which can further benefit the following researches such as learning dynamic systems, observing specific patterns in the data stream, and performing comparisons on different sequences. We will add these comments in the revised manuscript.
>
> Additionally, in the revised manuscript, we will provide the quantitative results that compare our proposed algorithm with the true order and other methods in some order-given datasets.
>
> 2. [Concerns on the One-Shot Learning Experiments]
>
> To clarify, given a labeled data, we do not select nearest neighbor data for it. Instead, we treat our proposed GMN as a generative model and then generate a sequence of data. Consider the 5-way (i.e., 5 classes) 1-shot (i.e., 1 labeled data per class) task; now we'll have 5 sequences for different categories. Next, we determine the class of unlabeled data based on the fitness within each sequence, which means we determine the class based on the highest generation probability (see Eq. (4)). On the other hand, all the other approaches are deterministic models, which are not able to generate data. Note that, we only have 1 labeled data per class at testing time.
>
> 3. [Nearest Neighbor as a strong baseline]
>
> As far as we know, there is not much prior work on discovering the order in an unordered dataset. Therefore, we consider Nearest Neighbor as a baseline method. We will avoid the "strong" word in the revised manuscript.
>
> 4. [Distance Metric Free]
>
> We do not intend to claim the negative log-likelihood of a Gaussian has nothing to do with Euclidean distance. We aim to propose an algorithm that can discover the order based on the Markov chain generation probability. This is compared to the Nearest Neighbor sorting, which requires a pre-defined distance metric. To avoid the confusion, we will rephrase distance-metric-free term in the revised manuscript.
>
> 5. [Concerns on Generative Model Experiments]
>
> We will rephrase the section to avoid confusion with conventional experiments in the generative model.
>
> Fig. 2 illustrates the advantage of using our proposed algorithm for searching next state. Our transition operator is trained to recover the order in the entire dataset, and thus it could significantly reduce the problem of being stuck in similar states. Note that this is all carried out under a  unified model. Therefore, we adopt Nearest Neighbor search as a baseline comparison. To provide more thorough experiments, we will also provide the suggested baseline "linearly extrapolating based on two most recent samples" in the revised manuscript.
>
> 6. [Concerns on the Hyper Parameters]
>
> Our proposed algorithm is not very sensitive to the choice of hyperparameters. First, the total number of data in various datasets are different. For example, MNIST, Horse, and MSR_SenseCam have 60,000, 328, and 362 instances, respectively. Second, we can feed the entire dataset into a batch when the total number of data is small. That is, we can have b = 328 and 362 for Horse and MSR_SenseCam dataset, respectively. And the corresponding overlaps between batches (i.e., b_overlap) would be 328 and 362. Please see Alg. 2 for more details.
>
> 7. [Concerns on Related Works]
>
> Although we do not focus on clustering, we will add the discussion with the suggested paper in the revised manuscript

---

### Official Review · AnonReviewer2 · 2017-11-28
**REVISED - still not convinced about the significance of the work. Substandard experiments and confusing. I leave my score as it is.     --  The proposed model has a very interesting motivation but the description is not clear. The authors do not explain the basics that strongly define the GMN. The experimental results are hard to follow since no intuition is provided.**

**Rating:** 4
**Confidence:** 4

**Review:**


The authors deal with the problem of implicit ordering in a dataset and the challenge of recovering it, i.e. when given a random dataset with no explicit ordering in the samples, the model is able to recover an ordering. They propose to learn a distance-metric-free model that assumes a Markov chain as the generative mechanism of the data and learns not only the transition matrix but also the optimal ordering of the observations.


> Abstract
“Aiming to find such orders, we introduce a novel Generative Markov Network (GMN) which we use to extract the order of data instances automatically. ”
I am not sure what automatically refers here to. Do the authors mean that the GMN model does not explicitly assume any ordering in the observed dataset? This needs to be better stated here.
“Aiming to find such orders, we introduce a novel Generative Markov Network (GMN) which we use to extract the order of data instances automatically; given an unordered dataset, it outputs the best -most possible- ordering.”

Most of the models assume an explicit ordering in the dataset and use it as an integral modelling assumption. Contrary to that they propose a model where no ordering assumption is made explicitly, but the model itself will recover it if any.

> Introduction
The introduction is fairly well structured and the example of the joint locations in different days helps the reader.

In the last paragraph of page 1, “we argue that … a temporal model can generate it.”, the authors present very good examples where ordered observations (ballerina poses, video frames) can be shuffled and then the proposed model can recover a temporal ordering out of them. What I would like to think also here is about an example where the recovered ordering will also be useful as such. An example where the recovered ordering will increase the importance of the inferred solution would be more interesting..



2. Related work
This whole section is not clear how it relates to the proposed model GMN. Rewriting is strongly suggested.
The authors mention Deep Generative models and One-shot learning methods as related work but the way this section is constructed makes it hard for the reader to see the relation. It is important that first the authors discuss the characteristics of GMN that makes it similar to Deep generative models and the one-shot learning models. They should briefly explain the characteristics of DGN and one-shot learning so that the readers see the relationship.
Also, the authors never mention that the architecture they propose is deep.

Regarding the last paragraph of page 2, “Our approach can be categorised … can be computed efficiently.”:
Not sure why the authors assume that the samples can be sampled from an unmixed chain. An unmixed chain can also result in observing data that do not exhibit the real underlying relationships. Also the authors mention couple of characteristics of the GMN but without really explaining them.  What are the explicit and implicit models [1] … this needs more details.

[1] P. J. Diggle and R. J. Gratton. Monte Carlo methods of inference for implicit statistical models. Journal of the Royal Statistical Society. Series B (Methodological), pages 193–227, 1984.

“Second, prior approaches were proposed based on the notion of denoising models. In other words, their goal was generating high-quality images; on the other hand, we aim at discovering orders in datasets.” —>this bit is confusing. Do the authors mean that prior approaches were considering the observed ordering as part of the model assumptions and were just focusing on the denoising?

3. Generative Markov models
First, I would like to draw the attention of the authors on the terminology they use. The states here are not the latent states usually referred in the literature of Markov chains. The states here are observed and should not be confused with the emissions also usually stated in the corresponding literature. There are as many states as the number of observations and not differentiation is made for ties. All these are based on my understanding of the model.

In  the Equation just before equation (1),  on the left hand side, shouldn’t \pi be after the `;’. It’s an average over the possible \pi.  We cannot  consider the average over \pi when we also want to find the optimal \pi.  The sum doesn’t need to be there. Shouldn’t it just be  max_{\theta, \pi} log P({s_i}^{n}_{i=1}; \pi, \theta) ?
Equation (1), same. The summation over the possible \pi is confusing. It’s an optimisation problem…

page 4, section 3.1: The discussion about the use of Neural Net for the construction of the transition matrix needs expansion. It is unclear how the matrix is constructed. Please add more details. E.g. use of soft-max non-linear transformation so that the output of the Neural Net can be interpreted as the probabilities of jumping to one of the possible states. In this fashion, we map the input (current state) and transform it to the probability gf occupying states at the next time step.

Why this needs expansion: The construction of the transition matrix is the one that actually plays the role of the distance metric in the related models. More specifically, the choice of the non-linear function that outputs the transition probability is crucial; e.g. a smooth function will output comparable transition probabilities to similar inputs (i.e. similar states).

section 3.2:
My concern about averaging over \pi applies on the equations here too.

“However, without further assumption on the structure of the transitional operator..”—> I think the choice of the nonlinear function in the output node of the NN is actually related to the transition matrix and defines the probabilities. It is a confusing statement to make and authors need to discuss more about it. After all, what is the driving force of the inference? This is a problem/task where the observations are considered in a number of different permutations. As such, the ordering is not fixed and the main driving force regarding the best choice of ordering should come from the architecture of the transition matrix; what kind of transitions does the Neural Net architecture favour? Distance free metric but still assumptions are made that favour specific transitions over others.

“At first, Alg. 1 enumerates all the possible states appearing in the first time step. For each of the following steps, it finds the next state by maximizing the transition probability at the current step, i.e., a local search to find the next state. ” —>  local search in the sense that the algorithm chooses as the next state the state with the biggest transition probability (to it) as defined in the Neural Net (transition operator) output? This is a deterministic step, right?

4.1 DISCOVERING ORDERS IN DATASETS
Nice description of the datasets. In the <MSR_SenseCam> the choice of one of the classes needs to be supported.  Why? What do the authors expect to happen if a number of instances from different classes are chosen?

4.1.1 IMPLICIT ORDERS IN DATASETS
The explanation of the inferred orderings for the GMN and Nearest Neighbour model is not clear. In figure 2, what forces the GMN to make distinguishable transitions as opposed to the Nearest neighbour approach that prefers to get stuck to similar states? Is it the transition matrix architecture as defined by the neural network?

>> Figure 10: why use of X here? Why not keep being consistent by using s?

*** DO the authors test the model performance on a ordered dataset (after shuffling it…) ?  Is the model able of recovering the order? **

---

> ### Author Response · Authors · 2017-12-22
> **Rebuttal**
>
> 1. [Concern on the Abstract]
> The term "automatically" refers to the meaning that our proposed GMN assumes this order can be learned even though it is not given explicitly. We will clarify this in the revised manuscript.
>
>
> 2. [Concern on the Introduction]
> Consider the task of studying evolutions for galaxy or star systems. Usually, the process takes millions or even billions of years, and it is infeasible for a human to collect successive data points manifesting meaningful changes. Therefore, we propose to recover the evolution when just providing a snapshot of thousands of data points. Similar arguments can be made in the study of slow-moving human diseases such as Parkinson's. On the opposite side, the cellular or molecular processes are too fast to permit entire trajectories. In these applications, scientists would like to recover the order from non-sequenced and individual data, which can further benefit the following researches such as learning dynamic systems, observing specific patterns in the data stream, and performing comparisons on different sequences. We will add these comments in the revised manuscript.
>
> 3. [Concern on Related Work]
> We thank the Reviewer for providing helpful suggestions for improving Related Work section. We will make more clear connections between our proposed GMN and Deep Generative Models as well as One-Shot Learning Models. Moreover, since we utilize deep neural networks for amortizing the large state space in the transitional operator, we consider our model as a deep model.
>
> All previous works build on a strong assumption that the chain needs to be mixed, while in practice it’s very hard to judge whether a chain is mixing or not. As a comparison, our model is free of this assumption, because the underlying model does not build on any property related to the stationary distribution. It is not our intent to claim that the unmixed chain can result in exhibiting real data relationships. We will clarify this as well as the differences between "implicit" and "explicit" model in the revised manuscript.
>
> Additionally, prior work proposed to learn the Markov chain such that the data are gradually denoised from low-quality to high-quality images. On the other hand, our model aims to order the data by assuming the order follows Markov chain data generation order.
>
> 4. [Concerns on the Generative Markov Models]
>
> Yes, we agree that it’s very important to describe in more detail on how to construct the transitional operators using neural networks. As the reviewer has pointed out, this essentially plays the role of the implicit distance metric in our model. We thank the reviewer for this suggestion and we will definitely expand the discussion in a revised version.  In the current version, we briefly discuss the neural network parametrization in Sec. 3.4. More specifically, we consider two distribution families (Bernoulli for binary-valued state and Gaussian for real-valued state). Also, this is a proper transitional operator. That is, sum of f(s,s') is 1.0. We use the conditional independence assumption which is also adopted in Restricted Boltzmann Machines. We will note this in the revised manuscript.
>
> 5. [Concerns on Sec. 4.1]
>
> We randomly partition the entire datasets into batches, which means that, in each batch, we do not assume all the classes are available nor an equal number of instances per class. We will clarify this in the revised manuscript.
>
> 6. [Concerns on Sec. 4.1.1]
>
> Fig. 2 illustrates the advantage of using our proposed algorithm for searching next state. Our transitional operator is trained to recover the order in the entire dataset, and thus it could significantly reduce the problem of stucking in similar states. The distinguishable transitions benefit from our algorithm instead of the architecture design for the transitional operator. However, the parametrization from Neural Network is also crucial. Neural Network serves as an universal function approximator, which enables us to amortize the large state space for every single state in a unified model.
>
> 7. [Concerns on the Consistency between x and s]
>
> We will unify the notation in the revised manuscript.
>
> 8. [Evaluation on Ordered Dataset]
>
> We do provide the evaluation with the ordered dataset (Moving MNIST) in Supplementary. In the revised manuscript, we will also provide the quantitative results that compare our proposed algorithm with the true order and other methods for more order-given datasets.

---

### Official Review · AnonReviewer3 · 2017-11-30
**This paper is well written and experiments are carefully done. However it is unclear how impactful are the results.**

**Rating:** 4
**Confidence:** 4

**Review:**

The paper is about learning the order of an unordered data sample via learning a Markov chain. The paper is well written, and experiments are carefully performed. The math appears correct and the algorithms are clearly stated. However, it really is unclear how impactful are the results.

Given that finding order is important, A high level question is that given a markov chain's markov property, why is it needed to estimate the entire sequence \pi star at all? Given that the RHS of the first equation in section 3.2 factorizes, why not simply estimate the best next state for every data s_i?

In the related works section, there are past generative models which deserve mentions: Deep Boltzmann Machines, Deep Belief Nets, Restricted Boltzmann Machines,  and Neural Autoregressive Density Estimators.

Equation 1, why is P(\pi) being multiplied with the probability of the sequence p({s_i}) ? are there other loss formulations here?

Alg 1, line 7, are there typos with the subscripts?

Section 3.1 make sure to note that f(s,s') sums to 1.0, else it is not a proper transition operator.

Section 3.4, the Bernoulli transition operators very much similar to RBMs, where z is the hidden layer, and there are a lot of literature related to MCMC with RBM models.

Due the complexity of the full problem, a lot of simplification are made and coordinate descent is used. However there are no guarantees to finding the optimal order and a local minimum is probably always reached. Imagining a situation where there are two distinct clusters of s_i, the initial transition operator just happen to jump to the other cluster. This would produce a very different learned order \pi compared to a transition operator which happen to be very local. Therefore, initialization of the transition operator is very important, and without any regularization, it's not clear what is the point of learning a locally optimal ordering.

Most of the ordering results are qualitative, it would be nice if a dataset with a ground truth ordering can be obtained and we have some quantitative measure. (such as the human pose joint tracking example given by the authors)

In summary, there are some serious concerns on the impact of this paper. However, this paper is well written and interesting.

---

> ### Author Response · Authors · 2017-12-21
> **Rebuttal**
>
> 1. [Impact of the Results]
>
> We think finding the implicit order in a given set is an important problem and the proposed method could be applied in various domains, including studying galaxy evolutions/human diseases, and recovering videos from image frames.
>
> 2. [Why estimating the entire sequence \pi?]
>
> In our algorithmic development we indeed estimate the best next state for each given state in the dataset (See Alg. 1, Line 5). But such greedy heuristics is a local search strategy and does not guarantee the globally optimal ordering that maximizes the likelihood function.
>
> On the other hand, we have also conducted experiments for estimating the best next state given every state s_i. Unfortunately, this makes the learned Markov chain stuck in a few dominant modes. To fix this, we treat the Markov chain generation process as the permutation (i.e., an implicit order) of the data. This modification encourages the state to explore different states without having the issue of collapsing into few dominant modes. We will clarify this in the revised manuscript.
>
> 3. [Permutation \pi]
>
> We assume the dataset exhibits an implicit order \pi^* which follows the generation process in a Markov chain. However, the direct computation is computationally intractable (i.e., the total number of data may be too large). In Sec. 3.3, we relax the learning of the order from the entire dataset into different batches of the dataset. To ensure an ergodic Markov chain, we assure the batches overlap with each other.
>
>
> 4. [Related Generative Models]
>
> We will add the discussions in related work for Deep Boltzmann Machines, Deep Belief Nets, Restricted Boltzmann Machines, and Neural Autoregressive Density Estimators.
>
> 5. [Typos and Clarifications]
>
> There is an additional term (a typo) \sum_{\pi \in \Pi(n)} in Eq. (1). However, the prior of permutation (i.e., \pi) may not be uniform, and thus P(\pi) should not be avoided in Eq. (1).
>
> There is also a typo in line 7, Alg. 1.
>
> We will fix these typos in the revised manuscript.
>
> 6.  [Transitional Operator]
>
> Sum of f(s,s') is 1.0. We use the conditional independence assumption which is also adopted in Restricted Boltzmann Machines. We will note this in the revised manuscript. Other MCMC approaches related to RBM will also be discussed in Sec. 3.4 in the revised manuscript.
>
> 7. [No guarantees to finding the optimal order]
>
> In the revised version we have shown that finding the globally optimal order in a given Markov chain and a dataset is NP-complete, hence there is no efficient algorithm that can find the optimal order. We argue that in this sense, locally optimal order obtained using greedy heuristics is favorable in many real-world applications.
>
> 8. [Concern on Initialization]
>
> We have tried three different initializations in our experiments. The first is to use Nearest Neighbor with Euclidean distance to suggest an initial order, and then train the transitional operator based on this order in few iterations (i.e., 5 iterations). The second is replacing Euclidean distance with L1-distance. The third is random initialization. We observe that even for the random initialization, the order recovered from our proposed algorithm still leads to a reasonable one that avoids unstable jumps between two distinct clusters. Therefore, we argue that the initialization may not be so crucial to our algorithm. We will add the discussion in the revised manuscript.
>
> 9. [Quantitative Results]
>
> In the revised manuscript, we will provide the quantitative results that compare our proposed algorithm with the true order and other methods for some order-given datasets.

---

### Decision · Program_Chairs · 2018-01-29
**ICLR 2018 Conference Acceptance Decision**

**Decision:**

Reject

**Comment:**

The problem of discovering ordering in an unordered dataset is quite interesting, and the authors have outlined a few potential applications. However, the reviewer consensus is that this draft is too preliminary for acceptance. The main issues were clarity, lack of quantitative results for the order discovery experiments, and missing references. The authors have not yet addressed these issues with a new draft, and therefore the reviewers have not changed their opinions.